# A-ADAPT: Adaptive Intracranial Artery Segmentation with Morphology-Guided Prompts and Difficulty-Aware Learning

**Zhiwei Tan**[1]                                    ZHIWEI24@UW.EDU
**Xin Wang**[1]                                        XWANG99@UW.EDU
**Meng Wang**[2]                              WANGMENG9218@126.COM
**Zixuan Liu**[3]                          ZUCKSLIU@CS.WASHINGTON.EDU
**Yin Guo**[4]                                          YINGUO@UW.EDU
**Jiamin Xia**[5]                                      JIAMINX@UW.EDU
**Niranjan Balu**[4]                                      NINJA@UW.EDU
**Linda Shapiro**[3]                        SHAPIRO@CS.WASHINGTON.EDU
**Chun Yuan**[4,6]                                       CYUAN@UW.EDU
**Mahmud Mossa-Basha**[7]                       MMOSSABASHA@UABMC.EDU

[1] *Department of Electrical and Computer Engineering, University of Washington, Seattle, United States*

[2] *Centre for Innovation and Precision Eye Health & Department of Ophthalmology, Yong Loo Lin School of Medicine, National University of Singapore, Singapore*

[3] *Paul G. Allen School of Computer Science and Engineering, University of Washington, Seattle, United States*

[4] *Department of Radiology, University of Washington, Seattle, United States*

[5] *Department of Bioengineering, University of Washington, Seattle, United States*

[6] *Department of Radiology and Imaging Sciences, University of Utah, Salt Lake City, United States*

[7] *Department of Radiology, University of Alabama at Birmingham, Alabama, United States*

**Editors:** Accepted for publication at MIDL 2026

## Abstract

Accurate segmentation of intracranial arteries in CTA and MRA is essential for cerebrovascular analysis but remains challenging due to fine-scale artery morphology, modality-dependent appearance, and frequent structural discontinuities. Existing CNN or Transformer based models struggle to generalize across modalities, while SAM-based methods rely heavily on manually provided prompts and often fail to preserve thin or low-contrast arteries. We propose A-ADAPT, an adaptive intracranial artery segmentation framework that enhances SAM with modality-aware representation learning, automatic morphology-guided prompting, and difficulty-aware optimization. First, a Cross-Modality Task Adapter (CMTA) aligns CTA and MRA feature distributions while preserving shared vascular characteristics. The Frequency Adapter (FA) and the Tubular Morphology Adapter(TMA) work together to refine artery representation by enhancing structural detail and highlighting the continuity of tubular anatomy. To eliminate dependence on manual prompts, we introduce an Automatic Directional Morphology Prompt Encoder (AutoDM-Prompt), which generates artery-aware prompts directly from the input image. Additionally, a difficulty-aware loss dynamically upweights uncertain or discontinuity-prone regions, enabling the model to better recover small branches and reduce false positives. Experiments on CTA and MRA datasets show that A-ADAPT achieves higher accuracy, and better structural continuity compared to several state-of-the-art methods. We release our code at: https://github.com/VV1111/A-ADAPT.

**Keywords:** CTA, MRA, Intracranial artery segmentation, automatic prompt, SAM

## 1. Introduction

Accurate segmentation of intracranial arteries in CT angiography (CTA) and MR angiography (MRA) is essential for cerebrovascular analysis but remains highlt challenging (Ko et al., 2017; Min et al., 2012; Orouskhani et al., 2024, 2025). Intracranial arteries are thin, tortuous, and extensively branching; small distal arteries often exhibit low contrast and are easily obscured by noise, calcification, or partial-volume effects. Moreover, the vascular tree possesses a sensitive topological structure, and errors in continuity or bifurcation geometry can substantially degrade downstream clinical analyses. The substantial differences in imaging physics and contrast mechanisms between CTA and MRA further exacerbate these challenges, making robust cross-modality artery segmentation particularly difficult.

Deep learning–based vascular segmentation has progressed substantially in recent years. CNN architectures such as U-Net(Ronneberger et al., 2015) variants and nnUNet (Isensee et al., 2021) remain widely used due to their strong multi-scale feature extraction, yet their inherently local receptive fields limit the modeling of long-range vessel trajectories. Transformer-based models alleviate this limitation by employing global self-attention to capture extended anatomical dependencies and improve continuity preservation. Beyond these general architectures, several vascular-specific enhancements have been proposed, including directional or anisotropic convolutions (e.g., DSCNet (Hu et al., 2023)) that highlight elongated structures, slice-to-slice fusion strategies (e.g., Deu-Net 2.0 (Dong et al., 2022)) that reduce discontinuities, and methods incorporating explicit geometric or topological constraints (e.g., GLCP (Zhou et al., 2025)) to mitigate branch breakage and false connections. However, these approaches often rely on modality- or task-specific priors, limiting their generalizability across datasets with different contrast or noise characteristics. Consequently, they often require dataset-specific retraining and tend to degrade when applied across modalities or institutions.

Beyond task-specific segmentation networks, the Segment Anything Model (SAM (Kirillov et al., 2023)), built upon a ViT encoder and promptable mask decoder, has demonstrated remarkable zero-shot segmentation capability in natural images through its prompt-driven paradigm—accepting points, bounding boxes, or coarse masks.This has sparked numerous adaptations in medical imaging (Wang et al., 2026). For example, Polyp-SAM (Li et al., 2024) fine-tunes SAM for colorectal polyp segmentation and achieves strong performance, but its model design remains tightly coupled to one organ and modality. AutoSAM (Shaharabany et al., 2023) introduces automatic prompt generation, reducing manual interaction; however, in complex and highly detailed medical images, its automatically generated prompts often fail to align precisely with fine structures such as small arteries. More general medical counterparts have emerged: MedSAM (Ma et al., 2024) and MedSegX (Zhang et al., 2025) fine-tune SAM or insert adapter layers to improve medical-domain specialization and establish multi-modality foundation segmentation model; I-MedSAM (Wei et al., 2024) introduces an implicit representation of masks conditioned on SAM features to better model anatomical shapes.

Despite recent progress, SAM and its medical variants remain limited for intracranial artery segmentation. They depend on explicit user prompts, which is impractical for volumetric CTA or MRA. Moreover, existing adaptations lack vascular structural priors, making small-branch recovery and continuity preservation difficult.

To overcome the challenges of multimodal intracranial artery segmentation, we introduce A-ADAPT, a structure-enhanced intracranial artery framework that integrates automatic prompting, multi-source structural adapters, and topology-aware optimization. Accordingly, A-ADAPT incorporates the following components. (1) **Automatic Directional Morphology Prompt Encoder** To address the impracticality of manual point or box prompts in 3D CTA/MRA segmentation, AutoDM-Prompt extracts multi-directional morphological responses and converts them into dense and sparse prompt embeddings that emphasize vascular trajectories, enabling fully automatic and vessel-informed segmentation. (2) **Tubular Morphology Adapter (TMA)** To better encode thin, low-contrast, and highly curved arteries in transformer-based models, TMA injects lightweight artery-enhancement responses into each ViT block, providing geometric reinforcement to fragile distal branches. By explicitly strengthening elongated structures that self-attention tends to underemphasize, TMA improves fine-branch recovery without added computational burden. (3) **Frequency Adapter (FA) and Cross-Modality Task Adapter (CMTA)** To reduce the appearance gap between CTA and MRA and stabilize feature representations, FA injects modality-robust spectral patterns into the patch space, mitigating contrast variability, while CMTA adapter further aligns encoder activations through modality embeddings. Together, they improve cross-modality robustness while preserving the structural detail required for accurate artery segmentation. (4) **Vascular Difficulty Aware Loss (VDLoss) for Topological Continuity** To prevent ambiguous or thin vascular regions to fracture during segmentation, VDLoss integrates prediction uncertainty and local boundary contrast to focus learning on structurally vulnerable regions, improving distal-artery recall and preserves the topological integrity of the arterial tree.

Our contributions are as follows: (1) We propose A-ADAPT to achieve robust and topology-preserving intracranial artery segmentation across CTA and MRA. (2) We introduce AutoDM-Prompt that eliminates manual prompting while providing explicit artery-aware guidance to the decoder. (3) Extensive experiments on CTA and MRA datasets demonstrate that our method achieves superior segmentation accuracy and better vascular continuity than existing approaches, including SAM variants, MedSAM, MedSegX, and nnUNet.

## 2. Method

To enable adaptive and structure-aware intracranial artery segmentation across CTA and MRA, we build upon the Segment Anything Model (SAM) and propose A-ADAPT, an adaptive intracranial artery framework that incorporates morphology-guided prompting and difficulty-aware learning. Figure 1 illustrates the overall architecture of A-ADAPT.

A-ADAPT enhances SAM's prompt-encoder-decoder backbone with three vascular-specific components. First, the artery adaptive image encoder enriches the ViT backbone through modality-conditioned and structure-sensitive feature representations. Second, the automatic directional morphology prompt encoder produces dense and sparse prompts directly from orientation-selective morphological responses, enabling geometry-preserving guidance without manual interaction. Third, the artery difficulty-aware loss directs optimization toward uncertain, thin, or structurally fragile vascular regions. The following sections describe each module in detail.

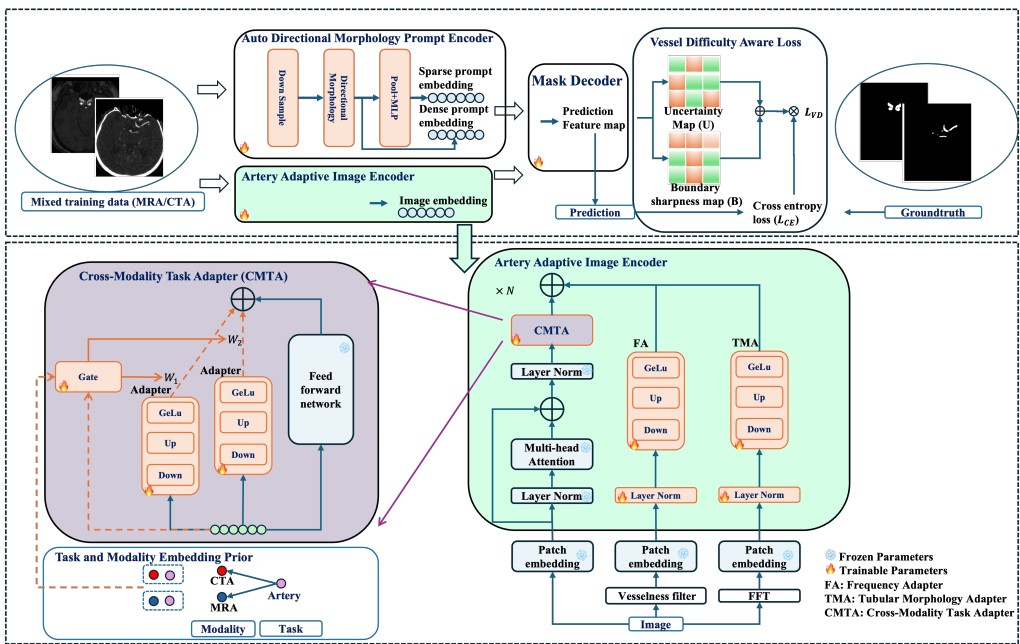

Figure 1: Overview of the proposed A-ADAPT framework. The model processes mixed CTA and MRA data through a cross-modality task adapter (CMTA) that adjusts feature transformation using modality and task embeddings. An automatic morphology prompt encoder extracts vessel cues from each image and converts them into prompt embeddings for the mask decoder. Together with artery-adaptive encoding and a difficulty-aware loss, the framework improves artery segmentation across heterogeneous appearances.

## 2.1. Artery Adaptive Image Encoder

To handle heterogeneous appearances and capture elongated, low-contrast vascular structures across CTA and MRA, we augment the standard ViT backbone with three complementary adapters: (1) a Cross-Modality Task Adapter that conditionally adjusts feature transformation based on CT/MRA characteristics; (2) a Tubular Morphology Adapter that introduces explicit geometric priors to better represent elongated vascular trajectories; (3) a Frequency Adapter that enhances artery-relevant frequency components and injects modality-robust spectral patterns. Together, these components inject structural and modality-aware priors into the encoder, enabling more robust learning of thin arteries, bifurcations, and regions prone to discontinuity.

**Cross-Modality Task Adapter (CMTA)**   The appearance of vascular structures varies significantly between CTA and MRA. To make the encoder aware of these differences, we introduce a **Task and Modality Embedding Prior**. The artery-segmentation task is represented by a learnable task embedding that is shared across all samples, while each input scan is tagged with a modality embedding indicating its modality. Together, these embeddings summarize what should be segmented and how the image typically appears. Both are learned jointly with the network and are injected into each CMTA block.

The CMTA module is inserted into every transformer block of the artery adaptive image encoder. Inside each CMTA, the incoming feature map first goes through several parallel

adapter branches (including Down, Up and GeLu), which provide multiple candidate refinements of the current representation. In parallel, the task embedding, and the modality embedding are combined with the current features. This combined vector is fed into a lightweight gating network that produces a set of mixing weights over the adapter branches. The final residual update is obtained by a weighted combination of these branches and added back to the feed-forward output.

**Tubular Morphology Adapter (TMA)**    Intracranial arteries appear as elongated tubular patterns that are easily fragmented or blurred under low contrast and noise. To explicitly enhance such structures, we introduce a TMA that integrates a arteryness-like response map into the encoder.

Given an input image $I$, we first compute a 2D tubular response map $T$ using a Hessian-based arteryness filter. Let $H(x)$ denote the Hessian matrix of $I$ at location $x$, and let $\lambda_1(x)$ and $\lambda_2(x)$ be its eigenvalues with $|\lambda_1(x)| \leq |\lambda_2(x)|$. A typical arteryness measure (Frangi et al., 1998) is defined as

$$V(x) = exp(-\frac{R_B(x)^2}{2\beta^2})(1 - exp(-\frac{S(x)^2}{2c^2})), \lambda_2(x) < 0, \tag{1}$$

where $R_B(x) = \dfrac{|\lambda_1(x)|}{|\lambda_2(x)| + \varepsilon}$, $S(x) = \sqrt{\lambda_1(x)^2 + \lambda_2(x)^2}$ and $\beta$, $c$ are fixed scale hyperparameters. In our implementation, we adapt a single-scale Hessian computation at the native image resolution, $H(x)$ is approximated using fixed $3 \times 3$ second-order finite-difference kernels. The resulting arteryness map $V$ is normalized and used as the Tubular Morphology map $T$.

To align $T$ with the transformer tokens, it is projected into the patch-embedding space and reused across all ViT blocks. Each block includes a lightweight tubular adapter (TMA) composed of a linear down-projection layer, a GELU activation layer, and a linear up-projection layer. The transformed tubular features are injected as an additive residual into the block output, strengthening the representation of thin and elongated arteries throughout the encoder. This single-scale design keeps the tubular morphology extraction lightweight and computationally efficient, aligning with our goal of minimally adapting SAM-based encoders.

**Frequency Adapter(FA)**    Vascular structures exhibit characteristic multi-scale frequency patterns, and the amplitude spectrum preserves these patterns more reliably than spatial domain intensities, particularly in low-contrast CTA and MRA. Motivated by this, we introduce a frequency adapter that integrates amplitude-spectrum information into the encoder.

Given an input image $I$, we compute its frequency representation using the Fast Fourier Transform (FFT). The spectrum at coordinate $(u, v)$ is

$$\mathcal{F}_{u,v} = \sum_{h=1}^{H} \sum_{w=1}^{W} I_{h,w} \, e^{-j2\pi\left(\frac{h}{H}u + \frac{w}{W}v\right)}. \tag{2}$$

From $\mathcal{F}_{u,v}$, we extract the amplitude spectrum $|\mathcal{F}_{u,v}|$, which empirically provides more stable structural representations than the phase component. After normalization, the amplitude

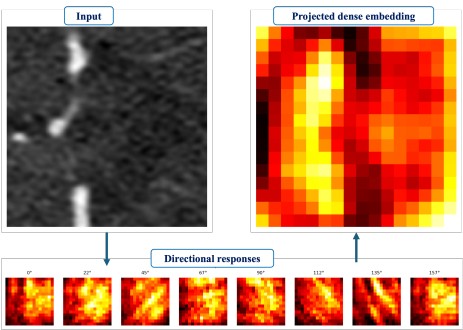

Figure 2: Overview of the Directional Morphology Block in AutoDM-Prompt. Top-left: the input; Bottom row: orientation-selective directional responses obtained across multiple angles; Top-right: dense embedding obtained by projecting the stacked responses via a $1 \times 1$ convolution. High-response regions (yellow) correspond to vessel trajectories and provide a dense prompt for the decoder.

map is projected into the patch-embedding space to align with the input tokens of the ViT blocks.

Each transformer block incorporates a lightweight frequency adapter that follows the same residual-injection principle as the tubular pathway. The frequency adapter provides a complementary global perspective by injecting spectral structural information throughout the encoder.

## 2.2. Automatic Directional Morphology Prompt Encoder

This section presents the Automatic Directional Morphology Prompt Encoder (AutoDM-Prompt), which generates artery-aware dense and sparse prompt embeddings directly from the input image. The goal is to replace manual point or box prompts with automatic, structure-preserving cues that better capture the tubular and orientation-dependent nature of vascular anatomy.

**Downsampling**    Given an input image $I \in \mathbb{R}^{B \times 3 \times H \times W}$, a lightweight convolutional backbone first produces a low-resolution feature map $F_0 \in \mathcal{R}^{B \times C_0 \times H_e \times W_e}$, where $(H_e, W_e)$ is the image embedding resolution and $C_0$ is a base channel. This feature map captures low-resolution but semantically meaningful appearance information.

**Directional Morphology Block**    To incorporate artery-specific geometric information, we introduce a Directional Morphology Block constructed from a bank of line-shaped kernels spanning multiple orientations between $0°$ and $180°$. Initialized to approximate soft morphological line filters and implemented as depthwise convolutions, these kernels extract orientation-selective responses. Each directional channel responds preferentially to artery segments aligned with its corresponding orientation, effectively capturing the elongated and anisotropic nature of vascular structures. Representative multi-directional activations are shown in Figure 2 (bottom).

The stacked directional responses are then projected onto the embedding dimension using a convolution $1 \times 1$ followed by normalization, which produces a dense embedding $F_{dense}$. As shown in Figure 2(top right), this embedding emphasizes continuous vascular

trajectories (yellow-highlighted regions), providing an effective dense prompt representation for the mask decoder.

In addition to the dense embedding, AutoDM-Prompt generates two sparse prompt tokens. The global average and global maximum pooling is applied to $F_{dense}$, and their pooled descriptors, which capture holistic structural trends and salient arterial responses, respectively, are passed through a lightweight MLP to produce two complementary vectors summarizing global vascular characteristics. These vectors $F_{sparse}$ serve as automatic sparse prompts, replacing manually supplied clicks in the standard SAM framework.

### 2.3. Vascular Difficulty Aware Loss

In this section, we discuss the proposed Vascular Difficulty Aware Loss (VDLoss), which is designed to emphasize training samples that contain thin branches, ambiguous boundaries, and potential artery discontinuities.

Let $z \in \mathbb{R}^{B \times 1 \times H \times W}$ be the logits predicted by the network for a batch of $B$ images. $\Omega = \{1, ..., H\} \times \{1, ..., W\}$ denotes the spatial domain of each image. For a pixel location $x \in \Omega$ in each image, we denote its logit by $z(x)$ and its foreground probability by $p(x) = \sigma(z(x)) \in [0, 1]$, where $\sigma(\cdot)$ is the sigmoid function.

Empirically, most topological errors in tubular structures, such as missing thin arteries or broken braches, occur in ambiguous regions where the model is not confident, or around sharp transitions between foreground and background. To capture these phenomena, we define two pixel-wise maps: an uncertainty map $U(x)$ (Kendall and Gal, 2017) and a boundary sharpness map $B(x)$ (Xie and Tu, 2015).

For each pixel, the uncertainty map $U(x)$ is defined as

$$U(x) = 4p(x)(1 - p(x)), \tag{3}$$

which reaches its maximum at $p(x) = 0.5$ and vanishes as $p(x)$ approaches 0 or 1. Therefore, high values of $U(x)$ indicate where the model is most uncertain about the artery label, which often corresponds to thin, low contrast branches and regions around potential breakpoints.

To characterize the strength of the boundary, we approximate the probability gradient using a $3 \times 3$ local contrast operator . Let $\mathcal{N}_3(x)$ denote the $3 \times 3$ neighborhood centered at $x$. We define

$$B(x) = \max_{y \in \mathcal{N}_3(x)} p(y) - \min_{y \in \mathcal{N}_3(x)} p(y). \tag{4}$$

A large value of $B(x)$ implies that the local patch contains a strong transition between background and artery. Such regions typically lie in artery boundaries, bifurcations, and abrupt prediction jumps that result in a topological break.

Based on the above two maps, we assign a difficulty weight $w(x)$ to each pixel:

$$w(x) = 1 + \alpha_u\, U(x) + \alpha_b\, B(x), \tag{5}$$

where $\alpha_u$ and $\alpha_b$ are hyper-parameters controlling the relative importance of uncertainty and boundary information, respectively. In order to avoid numerical instability and excessively large gradients, we clip the weight $w(x)$ into a bounded interval of $[1, 3]$.

Images containing many thin or fragmented arteries naturally deserve more attention. We compute a sample-level hardness score $h_b = \frac{1}{|\Omega|} \sum_{x \in \Omega} w(x)$ by averaging pixel-wise

weights, and normalize hardness by the mini-batch mean: $\hat{h}_b = \frac{h_b}{\frac{1}{B}\sum_{i=1}^{B} h_i + \varepsilon}$, which keeps the average hardness close to one and stabilizes the loss magnitude while preserving the relative difficulty across samples. Then the proposed VDloss reweights the cross entropy loss $\mathcal{L}_{CE}$ of each sample according to its normalized hardness $\mathcal{L}_{VD} = \frac{1}{B}\sum_{b=1}^{B} \hat{h}_b \mathcal{L}_{CE}$.

## 3. Results and Discussion

**Dataset and Experimental Setup.** We evaluate A-ADAPT on the TopCoW (Yang et al., 2025) dataset, which contains 125 paired CTA and MRA volumes. All experiments follow a 3D case-level split, where the cases are randomly partitioned into 80% for training, 10% for validation, and 10% for testing. Although the split is defined at the 3D volume level, the proposed model and all SAM-based baselines are trained in a 2D slice-wise manner. During training, CTA and MRA slices are sampled jointly and the models are optimized using AdamW (Loshchilov and Hutter, 2017) with a cosine learning rate schedule.

Unless otherwise specified, models are trained for 100 epochs with a batch size of 16, an initial learning rate of $1 \times 10^{-4}$, and weight decay of $1 \times 10^{-2}$. For inference, slice-wise predictions are aggregated to reconstruct full 3D vessel volumes.

**Compared Methods and Evaluation Metrics** We compare A-ADAPT with several representative segmentation approaches, covering both 3D volumetric architectures and SAM-based frameworks. As a 3D baseline, we include DSCNet(Hu et al., 2023), a convolutional network tailored for vascular segmentation , as well as SwinUNETR (Cao et al., 2022), a representative 3D transformer-based architecture widely used in medical image segmentation. For nnUNet(Isensee et al., 2021), we evaluate both its 2D and 3D configurations and report the variant that performs better for each modality. In addition, we benchmark four widely used SAM-derived medical segmentation models, namely MedSAM(Ma et al., 2024), MedSAM2(Ma et al., 2025), MedSegX(Zhang et al., 2025), and I-MedSAM(Wei et al., 2024), which serve as strong segmentation baselines adapted from the Segment Anything framework. All of these models standardize their prompt input using the ground-truth artery bounding box. For MedSAM2(Ma et al., 2025), during inference we use the bounding box extracted from the middle slice of each volume as the prompt, which serves as the initialization slice for its forward–backward slice propagation mechanism. Performance is measured using the Dice Similarity Coefficient (Dice) and Hausdorff Distance (HD) (Taha and Hanbury, 2015), which together assess volumetric overlap and boundary accuracy.

### 3.1. Quantitative Evaluation

Table 1 reports the quantitative results on the dataset. Across all methods, segmentation performance on MRA is generally higher than on CTA. This is likely because the high-contrast bone structures in CTA introduce strong intensity interference around arterial boundaries, making vessel delineation more difficult than in MRA. In MRA, A-ADAPT obtains the highest Dice of 95.62% and a competitive Hausdorff distance of 9.87. At the same time, the others also show relatively strong performance in MRA, benefiting from the more consistent and clearer contrast of the background of the vessel. In CTA, A-ADAPT also achieves the best Dice of 92.02% and Hausdorff distance 13.49, outperforming Med-SAM, MedSegX, I-MedSAM, DSCNet, nnUNet, as well as Swin-UNETR by noticeable

Table 1: Quantitative comparison on CTA and MRA, categorized by method type.

| Method | CTA | | MRA | |
|---|---|---|---|---|
| | Dice ($\uparrow$) | HD ($\downarrow$ voxel) | Dice ($\uparrow$) | HD ($\downarrow$ voxel) |
| *Prompt-based Methods* | | | | |
| MedSAM | $84.10 \pm 3.86$ | $17.35 \pm 6.26$ | $91.50 \pm 1.96$ | $10.44 \pm 5.23$ |
| MedSAM2 | $73.35 \pm 9.11$ | $13.58 \pm 4.79$ | $85.81 \pm 9.93$ | $12.46 \pm 6.37$ |
| MedSegX | $84.63 \pm 4.39$ | $19.05 \pm 7.40$ | $92.26 \pm 2.17$ | $9.92 \pm 4.62$ |
| I-MedSAM | $86.24 \pm 2.48$ | $20.63 \pm 9.65$ | $93.29 \pm 1.61$ | $12.88 \pm 4.28$ |
| *Prompt-free Methods* | | | | |
| nnUNet | $89.64 \pm 4.19$ | $16.78 \pm 6.29$ | $91.20 \pm 3.51$ | $11.94 \pm 7.01$ |
| SwinUnetr | $87.62 \pm 6.76$ | $15.89 \pm 4.01$ | $91.16 \pm 7.32$ | $10.56 \pm 3.27$ |
| DSCNet | $90.93 \pm 4.16$ | $19.59 \pm 9.33$ | $87.14 \pm 5.24$ | $15.85 \pm 6.04$ |
| Proposed | $\mathbf{92.02 \pm 3.52}$ | $\mathbf{13.49 \pm 7.00}$ | $\mathbf{95.62 \pm 2.47}$ | $\mathbf{9.87 \pm 6.04}$ |

margins. In contrast, SwinUnetr exhibits inferior performance in CTA, which we attribute to the limited number of annotated 3D training volumes and the increased appearance variability caused by bone-induced clutter. For MedSAM2, although it leverages slice-wise propagation initialized from a single prompt, its performance remains lower than other SAM-based methods, especially on CTA, indicating that prompt propagation alone is insufficient to address complex background interference. In general, the superior performance across modalities demonstrates the robustness of A-ADAPT in handling the heterogeneous vascular appearance.

### 3.2. Qualitative Evaluation

Figure 3 presents a visual comparison of segmentation results produced by different methods on representative CTA and MRA cases. For each case, both 3D reconstructions and 2D slices are shown to highlight branching continuity and boundary accuracy. There are some common failure patterns are observed across all results from the baselines. These include missed distal branches and discontinuities in thin or low-contrast arteries, as well as false-positive responses in bone, soft-tissue edges, or background artifacts. These issues are particularly evident for volumetric models and prompt-based methods on CTA, where complex anatomical clutter significantly affects segmentation consistency.

In contrast, A-ADAPT recovers more complete arterial trajectories and exhibits fewer mis-segmented structures. The morphology-guided directional prompt encoder enhances the model's ability to follow elongated vascular paths, while the frequency-enhanced representation improves robustness to contrast variation. Furthermore, the difficulty-aware optimization directs learning toward ambiguous or topology-sensitive regions, helping maintain branch continuity. These qualitative observations align well with the quantitative gains seen in Table 1.

### 3.3. Ablation Experiments

Table 2 summarizes the ablation experiments conducted with MedSAM as the baseline. When introducing the CMTA, the model begins to simultaneously leverage modality-specific

| Image | Label | nnUNet | DSCNet | SwinUnetr | MedSAM | MedSAM2 | MedSegX | I-MedSAM | Proposed |

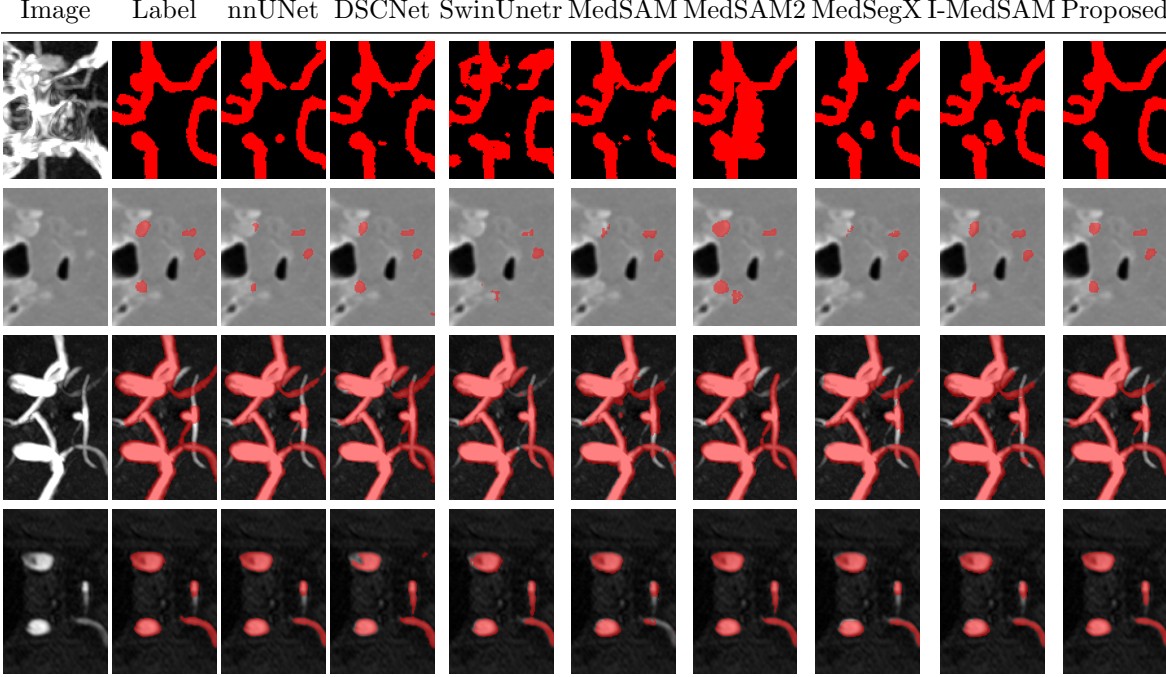

Figure 3: Visual comparison of different segmentation methods on CTA (rows 2–3) and MRA (rows 4–5). Rows 2 and 4 show MIP visualizations. Due to bone-induced clutter in CTA, we only display the predicted masks, whereas for MRA, predictions are overlaid on the original MIP. Rows 3 and 5 show representative slices highlighting local vessel structures.

Table 2: Ablation experiments of A-ADAPT on CTA and MRA.

| Modules | | | | | CTA | | MRA | |
| CMTA | FA | TMA | AutoDM | Loss | Dice ($\uparrow$) | HD ($\downarrow$,voxel) | Dice ($\uparrow$) | HD ($\downarrow$,voxel) |
|---|---|---|---|---|---|---|---|---|
| – | – | – | – | DiceBCELoss | $84.10 \pm 3.86$ | $17.35 \pm 6.26$ | $91.50 \pm 1.96$ | $10.44 \pm 5.23$ |
| ✓ | – | – | – | DiceBCELoss | $84.65 \pm 4.55$ | $18.26 \pm 7.34$ | $92.14 \pm 2.23$ | $\mathbf{9.46 \pm 4.41}$ |
| ✓ | ✓ | – | – | DiceBCELoss | $87.07 \pm 5.29$ | $15.03 \pm 6.24$ | $91.97 \pm 1.93$ | $9.72 \pm 4.26$ |
| ✓ | ✓ | ✓ | – | DiceBCELoss | $90.89 \pm 8.38$ | $14.36 \pm 10.25$ | $93.94 \pm 3.54$ | $10.50 \pm 5.93$ |
| ✓ | ✓ | ✓ | ✓ | DiceBCELoss | $91.86 \pm 4.29$ | $14.62 \pm 8.23$ | $94.72 \pm 2.36$ | $10.19 \pm 6.63$ |
| ✓ | ✓ | ✓ | ✓ | Dice+clLoss | $91.92 \pm 3.79$ | $13.97 \pm 9.13$ | $94.85 \pm 3.25$ | $10.06 \pm 7.22$ |
| ✓ | ✓ | ✓ | ✓ | VDLoss | $\mathbf{92.02 \pm 3.52}$ | $\mathbf{13.49 \pm 7.00}$ | $\mathbf{95.62 \pm 2.47}$ | $9.87 \pm 6.04$ |

appearance differences and shared vascular characteristics, resulting in consistent performance gains across both CTA and MRA. Incorporating the FA and TMA further strengthens the model's ability to detect thin and low-contrast arteries by improving boundary discrimination and structural continuity. For MRA, the inclusion of FA leads to a slight decrease in Dice while improving HD performance, reflecting a trade-off between voxel-wise overlap and global structural consistency in low-contrast scenarios.

Once AutoDM-Prompt is added, the decoder benefits from orientation-aware morphological guidance, which leads to more coherent branch reconstruction even in challenging anatomical regions. Finally, the VDLoss provides the most pronounced improvement by prioritizing uncertain and boundary-rich samples during optimization, effectively reducing the topological breaks that baseline models often encounter.The progressive improvements

Table 3: Computational overhead of different module combinations.

| Method | Trainable Params (M) | Total Params (M) | Training Epoch Time (s) | Peak GPU (MB) | MACs (G) |
|---|---|---|---|---|---|
| Base (MedSAM) | 5.37 | 92.25 | 60.5 | 6487 | 0.0105 |
| Base + CMTA | 13.84 | 93.62 | 67.0 | 25212 | 0.0106 |
| CMTA + FA | 13.85 | 93.62 | 68.1 | 25548 | 0.0107 |
| CMTA + FA + TMA | 13.86 | 93.62 | 68.5 | 25671 | 0.0107 |
| CMTA + FA + TMA + AutoDM | 14.28 | 94.05 | 70.4 | 26299 | 0.0123 |
| CMTA + FA + TMA + Dice+clLoss | 14.28 | 94.05 | 95.8 | 29352 | 0.0123 |
| CMTA + FA + TMA + VDLoss | 14.28 | 94.05 | 81.7 | 27352 | 0.0123 |

in Table 2 confirm that each component contributes distinct and complementary gains. To further assess the practical cost of these improvements, Table 3 reports the corresponding computational overhead for the same ablation settings. Despite the introduction of multiple artery-aware modules, the increase in trainable parameters and training time remains moderate relative to the achieved performance gains, indicating a favorable accuracy–efficiency trade-off.

## 4. Conclusions

In this paper, we introduced A-ADAPT, an adaptive intracranial artery segmentation framework that integrates modality-adaptive and structural enhancement encoding, automatic morphology-guided prompting, and difficulty-aware learning. By explicitly emphasizing thin vessel structures and topological continuity, A-ADAPT achieves state-of-the-art performance on both CTA and MRA in the TopCoW dataset.

In particular, our results also highlight an important practical observation: current 2D-based frameworks demonstrate strong capability to capture fine-grained local vascular features, benefiting from efficient training and relatively easier data collection. In contrast, existing 3D training objectives primarily emphasize global contextual understanding, while effectively balancing local detail preservation and global structural consistency remains challenging. Since accurate vessel segmentation critically depends on fine-grained local continuity as well as coherent global topology, this gap partly explains the competitive advantage of advanced 2D and hybrid approaches in the current setting.

Although A-ADAPT demonstrates strong robustness in two modalities, it has not yet been fully evaluated in other vascular territories or tubular anatomical structures. Furthermore, while the proposed framework implicitly promotes 3D structural continuity through boundary-aware and uncertainty-guided optimization, we believe that more effective and efficient 3D-aware training objectives, as well as explicit topology-aware analyzes such as connectivity-based statistics or structure-level consistency measures, could further improve segmentation performance and strengthen clinical relevance. Exploring such unified 3D formulations in additional imaging modalities and anatomical regions constitutes an important direction for future work.

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
