# OpenReview forum: "A-ADAPT: Adaptive Intracranial Artery Segmentation with Morphology-Guided Prompts and Difficulty-Aware Learning"
_MIDL.io/2026/Conference — MIDL 2026 Poster_

### Official Review · Reviewer_kzxt · 2026-01-07

**Confidence:** 4
**Preliminary Rating:** 4
**Final Rating:** 5

**Summary:**

Authors propose an architecture for MRA + CTA intracranial artery segmentation. While the architecture is based on automating SAM, the specific ideas for automation and modality specific improvements are interesting. The authors add many adapters to SAM that take into consideration specific features of the vessel segmentation problem.

**Strengths:**

I want to highlight the employment of vesselness filters, and frequency responses as appropriate enhancements of input features, adequate for the problem of vessel segmentation problems. The prompt automationg with line morphology is another interesting part of the adaptations for this specific problem. Balancing loss signals with uncertainty and boundary information also makes sense intuitively. Since ablations showcase improvements, these would be interesting findings for others in the community working with similar problems. Authors also promise public code in the future. In general the paper is well written.

**Weaknesses:**

My only perceived, possible weakness, are topics not addressed in the paper that are present in my questions below, and could enrich the discussion sections on the paper. Altough results are presented with quick discussions, I believe the paper lacks a proper discussion section going beyond the fact the proposed adapters seem to work.

**Detailed Comments:**

Minor comments:

* Figure 1 could be improved, in my opinion the authors used too much dotted lines.

* Its not clear how "up" and "down" are performed in the methodology.

**Justification Of Final Rating:**

Thank you for addressing my comments. Reporting MACS (multiply add count) in the computational overhead table would make it more complete and useful.  As the authors have addressed all my concerns in their response, I have improved my rating and believe this work should be accepted.

**Justification Of The Preliminary Rating:**

* In general, the paper presents interesting architecture modifications that are shown with ablations and comparisons to improve performance on the specific CTA/MRA vessel segmentation task, and is of interest to the community in MIDL. My initial recommendation is acceptance.

**Questions To Address In The Rebuttal:**

* Could the authors comment on the computational overhead introduced by CMTA, FA, TMA, and AutoDM-Prompt relative to vanilla SAM or MedSAM?

* Are Dice and Hausdorff Distance computed volumetrically on the reconstructed 3D volumes for all models?

* Why were competing 3D SAM based methods not explored? Example: SAM-Med3D

* On the same vein, could you justify why you chose to pursue a 2D architecture? Could a 3D variation achieve even better results?

* One question is how these modifications help with SAM's problem of excessive upsampling limiting detail on output maps smaller than the visual feature space, i.e, considering patch embeddings computed from 16x16, SAM tends to struggle with pixel level details. However from the results, your architecture, while based on SAM appears to surpass that problem. Could you elaborate?

---

> ### Author Response · Authors · 2026-01-25
>
> 1.improved Figure1
>
> Thank you for the suggestion. We apologize for the lack of clarity in the original Figure 1. In the revised manuscript, we have redrawn Figure 1 to improve readability by removing unnecessary dotted lines wherever possible.
> In addition, we explicitly indicate the “up” and “down” operations using directional arrows, and when illustrating individual modules in detail, we use consistent fill colors to denote paired components. These revisions aim to make the architecture flow and the corresponding operations clearer to the reader.
>
> 2.computational overhead
>
> Thank you for the comment. We quantitatively analyzed the computational overhead introduced by all the modules. As summarized in Table 3, the trainable parameters increase from 5.37M to 14.28M, while the total parameters remain nearly unchanged due to the frozen backbone.
> The training time per epoch increases from 60.5 s to 95.8 s, accompanied by a moderate rise in peak GPU memory usage. Overall, these additions remain lightweight and provide a favorable trade-off between computational cost and performance gains.
>
> 3.Are Dice and Hausdorff Distance computed volumetrically on the reconstructed 3D volumes for all models
>
> Yes. Both Dice and Hausdorff Distance are computed volumetrically on the reconstructed 3D volumes for all evaluated models.
>
> 4.Why were competing 3D SAM based methods not explored
>
> Thank you for the question. We did consider extending SAM to fully 3D settings and explored existing 3D SAM-based variants in preliminary experiments. However, we found that current 3D SAM-based approaches tend to struggle with thin, tubular structures such as intracranial vessels, and their performance is highly dependent on the availability of effective 3D prompts.
> In particular, at the time of this study, we did not identify a robust mechanism for automatic prompt generation in 3D volumes that could reliably capture vascular morphology. This limitation was one of the  key motivations for our work, which focuses on automatic, anatomy-aware prompt learning in a 2D setting.
> Following your suggestion, we have additionally included experiments with MedSAM2, and the corresponding results are reported in Table 1 and Figure 3.
>
> 5.Could a 3D variation achieve even better results
>
> We chose a 2D architecture because current 2D frameworks are effective at capturing fine-grained local features that are critical for vessel segmentation, while also benefiting from more stable optimization and easier data collection. In contrast, existing 3D training objectives often emphasize global context and face challenges in jointly preserving local detail and structural continuity, particularly for thin vascular structures.
> We agree that an ideal 3D-aware framework that effectively balances local feature fidelity and global topology could potentially achieve even better performance. Exploring such unified and efficient 3D formulations, including automatic prompt learning in 3D, is an important direction for future work and has been explicitly discussed in the revised manuscript.
>
>
> 6.SAM's problem of excessive upsampling limiting detail on output maps smaller than the visual feature space
>
> We agree that the coarse 16×16 patch embedding in SAM, together with heavy upsampling, can limit pixel-level detail recovery, particularly for thin vascular structures. Rather than modifying the patch resolution or decoder, our approach mitigates this issue by changing how fine-grained structural information is introduced and enforced.
> Specifically, vessel-aware morphological and directional cues are injected at the feature level, guiding the decoder toward structure-consistent predictions instead of relying on blind upsampling. In addition, the optimization objective emphasizes boundary-sensitive and uncertain regions, which discourages fragmented or over-smoothed vessel predictions.
> Together, these design choices shift the focus from recovering exact pixel detail to preserving anatomically meaningful structure, enabling improved delineation of thin vessels despite the coarse visual feature grid.

---

> > ### Comment · Reviewer_kzxt · 2026-01-26
> >
> > Thank you for addressing my comments. For visibility, I am copying here a small additional request: Reporting MACS (multiply add count) in the computational overhead table would make it more complete and useful.

---

> > > ### Author Response · Authors · 2026-01-27
> > >
> > > Thank you for the suggestion. Following this comment, we have added MACs to the computational overhead analysis.
> > >
> > > The updated results are summarized below.
> > >
> > > | Method | Trainable Params (M) | Total Params (M) | Training Epoch Time (s) | Peak GPU (MB) | MACs (G) |
> > > |---|---:|---:|---:|---:|---:|
> > > | Base (MedSAM) | 5.37 | 92.25 | 60.5 | 6487 | 0.0105 |
> > > | Base + CMTA | 13.84 | 93.62 | 67.0 | 25212 | 0.0106 |
> > > | CMTA + FA | 13.85 | 93.62 | 68.1 | 25548 | 0.0107 |
> > > | CMTA + FA + TMA | 13.86 | 93.62 | 68.5 | 25671 | 0.0107 |
> > > | CMTA + FA + TMA + AutoDM | 14.28 | 94.05 | 70.4 | 26299 | 0.0123 |
> > > | CMTA + FA + TMA + Dice+clLoss | 14.28 | 94.05 | 95.8 | 29352 | 0.0123 |
> > > | CMTA + FA + TMA + VDLoss | 14.28 | 94.05 | 81.7 | 27352 | 0.0123 |

---

### Official Review · Reviewer_d4To · 2026-01-08

**Confidence:** 3
**Preliminary Rating:** 3
**Final Rating:** 4

**Summary:**

This paper proposes A-ADAPT, an adaptive framework for intracranial artery segmentation in CTA and MRA that extends SAM with morphology-guided prompting, modality-aware adapters, and difficulty-aware optimization. The method introduces several components, including a Cross-Modality Task Adapter, Frequency and Tubular Morphology Adapters, an Automatic Directional Morphology Prompt Encoder to eliminate manual prompts, and a Vascular Difficulty-Aware Loss to preserve vessel continuity. Experiments on the public TopCoW dataset demonstrate improved Dice scores and reduced Hausdorff distance compared to SAM-based methods and strong CNN baselines such as nnU-Net. Overall, the paper targets an important and challenging clinical problem and shows convincing quantitative improvements, though at the cost of considerable architectural complexity.

**Strengths:**

- The paper addresses a highly relevant and difficult clinical task, namely intracranial artery segmentation across CTA and MRA, where thin structures, discontinuities, and modality gaps are well-known challenges.
- The proposed framework introduces multiple vascular-specific innovations, including morphology-guided automatic prompting and topology-aware loss design.
- The  paper provides comparisons with non-SAM baselines.

**Weaknesses:**

- The method is quite complex and introduces multiple interacting components,introducing multiple adapters (CMTA, FA, TMA), a custom prompt encoder, and a novel loss, all layered on top of SAM. While each component is motivated, the overall system risks being difficult to reproduce, maintain, and generalize.

- The paper relies on 2D slice-wise training for a  3D  problem, with limited discussion of how slice-wise inconsistencies affect global vessel connectivity.

**Detailed Comments:**

- The authors may consider simplifying or consolidating adapters, or providing guidance on which components are essential in practice.

- A discussion of runtime, memory overhead, and training stability would be valuable given the number of added modules.

- More explicit analysis of 3D topological consistency  would strengthen the clinical relevance.

**Justification Of Final Rating:**

Thank you for addressing all my comments and concerns. After reviewing the authors’ detailed responses and revisions, I am satisfied that my issues have been resolved, and I have therefore updated my rating to Weak Accept.

**Justification Of The Preliminary Rating:**

This paper presents a strong approach to a challenging segmentation problem, with convincing improvements over both SAM-based and CNN-based baselines. However, the contribution relies on a large number of interacting components, many of which build upon established ideas, resulting in a complex and potentially over-engineered system. While the empirical results are solid, the lack of external validation and the limited discussion of efficiency and generalization raise concerns about practical impact.

**Questions To Address In The Rebuttal:**

- Which components of A-ADAPT are strictly necessary to achieve most of the performance gains?
- How sensitive is the automatic prompt encoder to noise or artifacts in CTA compared to MRA?
-  What is the computational overhead introduced by the multiple adapters?

---

> ### Author Response · Authors · 2026-01-25
>
> 1.consider simplifying or consolidating adapters, or providing guidance on which components are essential in practice
>
> Thank you for the comment. Our goal is to build a unified framework that is robust across both CTA and MRA, rather than optimizing for a single modality or failure case. Under this goal, the proposed adapters are designed to address complementary limitations of directly applying SAM to vascular segmentation, and are already functionally decoupled rather than redundant.
>
> From a practical perspective, not all components are strictly required in every setting. The automatic prompt encoder and artery-adaptive image encoder form the core configuration, providing anatomy-aware guidance and vascular-specific representation. The difficulty-aware loss serves as an optional enhancement that further improves performance in challenging cases involving thin branches or ambiguous boundaries.
>
> Therefore, simplification does not necessarily require consolidating adapters, but can be achieved by selecting appropriate subsets of components based on data quality, computational budget, and application needs. The framework is intentionally modular to support such flexible deployment in practice.
>
> 2.relies on 2D slice-wise training for a 3D problem, with limited discussion of how slice-wise inconsistencies affect global vessel connectivity
>
> Thank you for the comment. The 2D method currently shows better performance on understanding finegrained local features such as vessels, due to more efficient training and easier data collection. On the other hand, the current 3D training objectives focus more on understanding the global pattern. While vessel structure needs fine-grained understanding of local features, Efficient 3D training objectives balancing local and global patterns are limited. Still, we agree that an ideal 3D aware training should lead to a better performance and leave this direction as a possible future work.  and we have explicitly added this discussion to the Conclusion section.
>
> 3.the computational overhead introduced by the multiple adapters
>
> Thank you for the comment. To address this, we quantitatively analyzed the computational overhead introduced by all the modules and the adapters. As summarized in Table 3, the trainable parameters increase from 5.37M to 14.28M, while the total parameters remain nearly unchanged due to the frozen backbone.
> The training time per epoch increases from 60.5 s to 95.8 s, accompanied by a moderate rise in peak GPU memory usage. Overall, these additions remain lightweight and provide a favorable trade-off between computational cost and performance gains.
>
> 4. More explicit analysis of 3D topological consistency
>
> Thank you for the insightful suggestion. We fully agree that more explicit analysis of 3D topological consistency would further strengthen the clinical relevance of this work. This has also been an important direction we are actively exploring, aiming to better align image-level representations with underlying vascular anatomy and medical interpretation.
> Motivated by your comment, we recognize that future 3D extensions of our framework could incorporate topology-specific quantitative measures, such as connectivity statistics, break or discontinuity analysis, and fractal-based structural descriptors, to provide a more explicit characterization of global vascular topology. We have added this discussion to the Conclusion and consider it a promising direction for future research.
>
> 5. sensitive is the automatic prompt encoder to noise or artifacts in CTA compared to MRA
>
> CTA and MRA differ substantially in their noise and artifact characteristics. CTA images often contain strong interference from bone structures and calcifications, which introduce sharp, high-frequency edges, whereas MRA generally presents smoother intensity distributions with fewer abrupt artifacts.
> In our framework, the automatic prompt encoder is not driven directly by raw intensity values but by feature representations that capture vessel morphology and orientation. This makes the prompt generation less sensitive to localized noise or spurious edges, particularly in CTA. In addition, the frequency adaptation and difficulty-aware optimization help reduce the influence of modality-specific artifacts by focusing learning on vessel-relevant patterns and uncertain boundary regions.
> In practice, this design leads to stable prompt generation across both modalities, with higher inherent robustness on MRA and effective mitigation of noise-related interference on CTA, as reflected by the consistent performance trends observed in our experiments.

---

### Official Review · Reviewer_mEL7 · 2026-01-09

**Confidence:** 4
**Preliminary Rating:** 4
**Final Rating:** 5

**Summary:**

The authors present A-ADAPT, a framework designed to adapt the Segment Anything Model (SAM) for intracranial artery segmentation in CTA and MRA. To address the limitations of manual prompting in 3D volumes, they introduce an AutoDM-Prompt encoder that extracts directional morphological cues. The model incorporates specific adapters for frequency (FA) and tubular morphology (TMA) to better represent thin, low-contrast vessels, alongside a cross-modality task adapter (CMTA). A difficulty-aware loss is utilized to focus the model on uncertain regions and boundaries. Testing on the TopCoW dataset shows A-ADAPT outperforms baselines like MedSAM and nnU-Net in both Dice and Hausdorff distance.

**Strengths:**

The transition from manual clicks to the AutoDM-Prompt system is a practical and necessary step for making SAM-based architectures viable for volumetric medical imaging. The use of Hessian-based priors and directional line filters is well-grounded in classical computer vision and effectively translates to vascular deep learning. The ablation study clearly demonstrates that the structural adapters (TMA and FA) provide significant gains, especially in CTA where bone interference is high. The cross-modality robustness is a strong point, as the model handles both CTA and MRA within a unified framework.

**Weaknesses:**

While the framework is trained in a 2D slice-wise manner, which may limit explicit modeling of full 3D vascular topology, the reported results suggest that the morphology-guided prompts and difficulty-aware optimization partially mitigate this issue in practice. The proposed difficulty-aware loss focuses on uncertain and boundary regions; however, a direct comparison with explicit topology-preserving losses (e.g., clDice) could further clarify its relative advantages. Additionally, the Hessian-based arteryness filter uses fixed parameters, and a multi-scale formulation may improve adaptability to varying vessel calibers. Finally, comparisons with recent 3D Transformer-based architectures (e.g., Swin-UNETR) would further strengthen the empirical evaluation.

**Detailed Comments:**

In Table 2, the MRA Dice does not increase monotonically with the addition of AutoDM; a brief explanation of this interaction or variance would improve clarity.

In Section 2.1, please clarify the specific range of scales used for the Hessian filter.

The clipping of weights $w(x) \in [1, 3]$ seems somewhat arbitrary; did the authors experiment with other ranges?

**Justification Of Final Rating:**

The authors have comprehensively addressed my concerns in the rebuttal. The inclusion of Swin-UNETR and MedSAM2 comparisons significantly strengthens the empirical evaluation. I appreciate the data-driven comparison between VDLoss and centerline-weighted losses; the evidence supports VDLoss as offering a superior efficiency-performance trade-off. While the 2D slice-wise limitation remains, the clarification on implicit boundary consistency is reasonable. Given the method's practical value in resolving the "prompting bottleneck" for volumetric segmentation and the rigorous additional validation provided, I believe this work is well-suited for publication.

**Justification Of The Preliminary Rating:**

The paper provides a technically sound extension of SAM to a high-impact clinical task. The novelty lies in the automated prompt generation and the integration of structural/frequency adapters that specifically target the tubular nature of arteries. While the 2D nature of the model is a slight limitation, the performance gains on the TopCoW benchmark are substantial enough to justify acceptance. It addresses the practical "prompting" bottleneck that often makes foundation models difficult to deploy in radiology workflows.

**Questions To Address In The Rebuttal:**

How does the model handle the transition between slices to ensure that a vessel doesn't "break" in the 3D reconstruction?

Did the authors compare the VDLoss against any topology-specific loss functions?

What is the added computational overhead of the AutoDM-Prompt and the adapters compared to the standard MedSAM?

---

> ### Author Response · Authors · 2026-01-25
> **We thank the reviewers for their constructive comments, which improved the clarity and completeness of the manuscript. In response, we expanded the ablation study to compare VDLoss with a topology-specific centerline-weighted loss, clarified key methodological details, added representative 3D baselines, and improved the overall presentation and writing of the manuscript.**
>
> 1.Did the authors compare the VDLoss against any topology-specific loss functions?
>
> We did a comparison between VDLoss and other topology-specific loss functions. Specifically, we explicitly compared VDLoss with a centerline-weighted loss by introducing a Dice + CLLoss setting in the ablation study. As shown in Table 2, incorporating the centerline-weighted loss consistently improves Dice and reduces Hausdorff Distance on both CTA and MRA compared to DiceBCE, demonstrating its effectiveness in enhancing vascular connectivity. However, the centerline-weighted loss introduces a substantially longer training time, as reported in Table 3, due to the additional centerline-related computations.
> In contrast, VDLoss achieves stronger performance gains while maintaining a more favorable balance between accuracy and efficiency, which motivates its selection as the final topology-aware optimization objective in our framework.
>
> 2.clarify the specific range of scales used for the Hessian filter
>
> The Hessian-based arteryness map used in the Tubular Morphology Adapter (TMA) is computed at a single implicit spatial scale, approximated by fixed 3×3 second-derivative finite-difference kernels applied at the native image resolution. We have clarified this implementation detail in the revised manuscript in the Tubular Morphology Adapter (TMA) (section 2.1.)
>
> 3.Comparison with Swin-UNETR
>
> In terms of 3D transformer-based comparisons, we have extended our experimental evaluation to include Swin-UNETR and MedSAM2, a recent SAM-based model with enhanced 3D generalization capability. These methods have been added to the benchmark in the revised manuscript.
> As shown in Table 1 and Figure 3, Swin-UNETR exhibits competitive performance on MRA but performs less favorably on CTA, which we attribute to the limited number of annotated 3D training volumes and the higher appearance variability in CTA. In contrast, its performance on MRA benefits from stronger structural consistency and lower inter-case variation.
>
> 4.slight non-monotonic variation in MRA Dice occurs
>
> Thank you for the comment. Regarding the non-monotonic Dice behavior on MRA, we note that the slight Dice decrease occurs specifically after introducing the Frequency Adapter (FA). This is because MRA already exhibits relatively homogeneous intensity patterns and clear vessel background contrast, where frequency-based enhancement provides limited additional benefit for voxel-wise overlap. In such cases, FA primarily acts as a regularizer that emphasizes global consistency rather than maximizing local overlap, which may lead to marginal Dice fluctuations while maintaining competitive Hausdorff Distance. Importantly, when FA is combined with topology-aware modules (TMA and AutoDM) and optimized with VDLoss, the complementary effects become evident, resulting in consistent improvements on MRA. We have clarified this interaction in the revised manuscript to improve interpretability.
>
> 5.weight clipping
>
> The clipping of w(x)to [1,3] is introduced to bound the influence of highly uncertain or sharp-boundary pixels. Since both uncertainty map U(x) and boundary sharpness map B(x)are bounded in [0,1], an unbounded w(x) could disproportionately amplify gradients when aggregated into the sample-level hardness​. The chosen range provides moderate emphasis on difficult regions while maintaining stable training dynamics after batch-level normalization.
>
> 6.model handle the transition between slices to ensure that a vessel doesn't "break" in the 3D reconstruction
>
> Thank you for the comment. The continuity of vascular structures across adjacent slices is primarily reflected by smooth intensity transitions and consistent boundary profiles in volumetric images. In our framework, this property is implicitly enforced during optimization rather than by explicit inter-slice post-processing. The proposed loss emphasizes boundary-consistent and uncertain regions, which aligns segmentation with smooth intensity transitions along vessels. This bias discourages abrupt slice-to-slice breaks and promotes coherent 3D reconstructions without explicit inter-slice tracking.
>
> 7.computational overhead
>
> Thank you for the comment. To address this, we quantitatively analyzed the computational overhead introduced by all the modules and the adapters relative to standard MedSAM. As summarized in Table 3, the trainable parameters increase from 5.37M to 14.28M, while the total parameters remain nearly unchanged due to the frozen backbone.
> The training time per epoch increases from 60.5 s to 95.8 s, accompanied by a moderate rise in peak GPU memory usage. Overall, these additions remain lightweight and provide a favorable trade-off between computational cost and performance gains.

---

### Author Rebuttal · Authors · 2026-01-25

**Rebuttal:**

We thank the reviewers for their insightful and constructive comments. Based on their feedback, we carried out additional analyses, clarifications, and revisions to both the experimental evaluation and the presentation of the manuscript. The main changes can be summarized as follows.

1. Comparative experiments
We extended the experimental evaluation by including additional representative baselines on MRA and CTA, namely Swin-UNETR and MedSAM2, to enable a broader comparison across 2D, 3D, and SAM-based approaches.

2. Ablation studies
We expanded the ablation study to directly compare VDLoss with a topology-specific centerline-weighted loss (Dice + CLLoss) on MRA and CTA. While the centerline-weighted loss improves connectivity-related metrics, it introduces substantially higher training cost.

3. Computational overhead
We quantitatively analyzed the computational overhead introduced by the proposed adapters. Although the number of trainable parameters and the training cost increase, the backbone remains frozen and the overall overhead is moderate. In practice, the framework is modular: the automatic prompt encoder and artery-adaptive image encoder constitute the core configuration, while the difficulty-aware loss can be optionally enabled depending on application needs and computational constraints.

4. Presentation and conceptual clarity
We improved the clarity of the manuscript by redrawing Figure 1, reducing unnecessary dotted lines, and explicitly indicating operations using clearer visual cues. We also clarified several methodological details, including the  Hessian filter and the volumetric computation of Dice and Hausdorff Distance. In addition, we expanded the discussion on 2D versus 3D modeling choices, slice-to-slice continuity, and future directions for explicit 3D topology-aware analysis in the revised Conclusion.

**Supporting Material:**

/attachment/5dd6e9a773f9081352924d14a9ac1b524e07a4ef.pdf

---

### Meta-Review · Area_Chair_MDTB · 2026-02-06

**Recommendation:** Accept (Oral)
**Confidence:** 5

**Metareview:**

All reviewers agree that this work tackles a challenging and clinically important problem of artery segmentation across CTA and MRA. The proposed approach is technically sound, well validated, and clearly outperforms existing methods. We recommend acceptance as an oral presentation.

---

### Decision · Program_Chairs · 2026-02-13

Accept (Poster)